# Rapid, Sensitive On-Site Detection of Deoxynivalenol in Cereals Using Portable and Reusable Evanescent Wave Optofluidic Immunosensor

**DOI:** 10.3390/ijerph19073759

**Published:** 2022-03-22

**Authors:** Yanping Liu, Yuyang Chen, Wenjuan Xu, Dan Song, Xiangzhi Han, Feng Long

**Affiliations:** 1School of Environment and Natural Resources, Renmin University of China, Beijing 100874, China; 2014101495@ruc.edu.cn (Y.L.); xuwenjuan@ruc.edu.cn (W.X.); sd@ruc.edu.cn (D.S.); 2016200727@ruc.edu.cn (X.H.); 2China National Intellectual Property Administration, Beijing 100088, China; chenyuyang@cnipa.gov.cn

**Keywords:** deoxynivalenol, evanescent wave, immunosensor, optofluidic, food detection

## Abstract

This paper develops an improved portable and reusable evanescent wave optofluidic immunosensor (OIP-v2) for rapid and sensitive on-site determination of deoxynivalenol (DON), one of the most frequently detected mycotoxins mainly produced by *Fusarium* species. Using the bifunctional reagent N,N′-Disuccinimidyl carbonate, deoxynivalenol-bovine-serum-albumin (DON-BSA) were covalently modified onto a bio-probe surface as biorecognition elements, whose robustness allowed it to perform multiple detections without significant activity loss. An indirect competitive immunoassay strategy was applied for DON detection. Under optimal conditions, the limit of detection of 0.11 μg/L and the linear dynamic detection range of 0.43 to 36.61 μg/L was obtained when the concentration of the Cy5.5-anti-DON antibody was 0.25 μg/mL. The OIP-v2 was also applied to detect DON in various cereals, and the recoveries ranged from 81% to 127%. The correlation between OIP-v2 and enzyme-linked immunosorbent assay (ELISA) through the simultaneous detection of maize-positive samples was in good agreement (R^2^ = 0.9891).

## 1. Introduction

Mycotoxins, a type of toxic secondary metabolites produced by fungi, have become a worldwide issue because they are commonly found in cereals (as well as in cereal-derived products) and can have serious toxic effects on health. Deoxynivalenol (DON), one of the most frequently detected mycotoxins mainly produced by *Fusarium* species, has attracted extensive attention due to its harmful influence on humans and animals [1]. DON can not only inhibit the synthesis of proteins and DNA and induce immuno-suppression, but can also result in gastroenteritis, vomiting, diarrhea, and even death [1,2,3]. Moreover, the heat resistance and water solubility of DON allow it to enter food chains and cause acute or chronic poisoning [4,5]. To reduce the daily intake of DON, numerous countries have set maximum allowable limits for DON residue in cereals and their derived products [6]. In China and the United States, the maximum permissible level of DON is 1 ppm in cereal and cereal products [7,8]. Traditional analytical methods for detecting DON include liquid chromatography-mass spectrometry [5], high-performance liquid chromatography, and matrix-assisted laser desorption/ionization time-of-flight mass spectrometry [6]. Although these methods are sensitive and accurate, they require expensive instruments, complex pre-treatment processes, and professionals, which are bottlenecks for the on-site detection of DON. Therefore, it is essential to explore low cost and reliable analytical technologies for the rapid and sensitive on-site detection of DON to protect human and animal health.

Biosensors are good alternatives for DON determination due to their high sensitivity, specificity, and rapidity. Various optical and electrochemical biosensors have been developed for DON determination [9,10,11,12,13,14,15,16]. Evanescent wave fluorescence biosensors, combining the advantages of evanescent wave fluorescence and biomolecules binding reactions, have been widely applied for the sensitive and rapid detection of pollutants. These biosensors generally use the evanescent wave, generating at the optical waveguide surface and decaying exponentially with distance, to excite the fluorescence labeled biorecognition molecules (e.g., antibody and aptamer) bound onto the biosensing surface. Because of the limited penetration depths (<100 nm) of the evanescent wave, the fluorescence-labeled antibodies bound onto the biosensing surface and the free fluorescence-labeled antibodies in the solution can be discriminated without a washed step. Quantitative detection can be achieved according to the linear relationship between the detected fluorescence intensity and the concentration of the target. Our group recently developed a portable platform (i.e., the evanescent wave optofluidic immunosensing platform, OIP) that combines the advantages of evanescent wave fluorescence, microfluidics, and immunoassay [17]. Compared with other evanescent wave fluorescence biosensors (e.g., RAPTOR [18] and Analyte 2000 [19]), the OIP had a simpler and more compact optical structure, in which no rigorous optical alignment was required. The OIP possesses great potential for detecting various pollutants because of its several unique characteristics including low reagent consumption, rapidity, reproducibility, and portability, which are also the desired characteristics for on-site detection of food pollutants.

Inspired by this, an improved evanescent wave optofluidic immunosensing platform (OIP-v2) was for the first time applied for the rapid and sensitive detection of DON in cereals. First, a novel modified method of the fiber optic bio-probe was proposed to improve its sensing performance in terms of stability and reusability. The hapten-carrier protein conjugates were covalently immobilized onto the bio-probe surface as biorecognition molecules using N,N′-disuccinimidyl carbonate (DSC) as the bifunctional reagent instead of other bifunctional reagents (e.g., glutaraldehyde). Second, a low-cost, Si-based photodiode detector (SOP-1000) with small volume and high sensitivity was used as an alternative to a photomultiplier tube (PMT) or lock-in amplifier for the detection of a weak fluorescence signal [18]. This not only reduced the cost instrument, but it also allowed for improved portability due to its small size. Third, a wireless communication module was added in the OIP-v2, enabling the detection results to be directly sent to the food quality management department, which provides key information for early-warning and pollution treatment if the concentration of pollutants exceeds the maximum value. Therefore, the OIP-v2 has great implications as a rapid accessible quantitative assay tool to satisfy the increasing demands of protecting civilians against possible direct exposure to pollution threats.

## 2. Materials and Methods

### 2.1. Materials and Reagents

N,N′-Disuccinimidyl carbonate (DSC), 3-Aminopropyl-triethoxysilane (APTS), N, N-Diisopropylethylamine (DIEA), bovine serum albumin (BSA), sodium dodecyl sulfate (SDS), ovalbumin (OVA), and DON were purchased from Sigma Aldrich (Shanghai, China). The anti-DON monoclonal antibody, DON-BSA, anti-enrofloxacin antibody (anti-ENR antibody), and anti-microcystin-LR (anti-MC-LR) antibody were purchased from the Shijiazhuang Zhiwei Biological Technology Co. (Hebei, China). Cy5.5 dye was purchased from GE Housecare (Buckinghamshire, UK). The Cy5.5 labeled anti-DON antibody (Cy5.5-anti-DON antibody) was prepared by our group. The standard solutions of DON were prepared by serial dilutions of the stored solution using a 0.01 M phosphate buffer solution (PBS). Corn and wheat were purchased from a local store in Beijing (China), and pig feed was purchased from Sanhui Feed (Beijing, China). Other reagents, if not specified, were purchased from Beijing Chemical Agents (Beijing, China). The SDS (pH = 1.9, 0.5%) was used as the regeneration solution.

### 2.2. Instrument: The Improved Evanescent Wave Optofluidic Immunosensing Platform (OIP-v2)

Figure 1 shows the scheme of OIP-v2 (the photo of OIP-v2 is shown Appendix A). The OIP-v2 has several unique characteristics (i.e., low reagent consumption, short assay time, enhanced immunosensing performance, and simplified platform design) that are beneficial for the on-site detection of food pollutants. Laser light (635 nm, 8.0 mW) is introduced into the fiber optic bio-probe using a single multi-mode fiber structure. The evanescent wave generates at the bio-probe surface when the incident light propagates along the length of the bio-probe via total internal reflection. The evanescent wave excites the surface-bound Cy5.5 labeled anti-DON antibody and causes excitation of the fluorophores, and part of fluorescence couples back into the bio-probe. The fluorescence collected by the single multi-mode fiber structure is detected by a Si-based photodiode detector after filtering using a bandpass filter (Edmund Optics, Shenzhou, China). The real-time fluorescence signal trace is recorded in a mini-computer. The whole DON process, including sample detection and bio-probe regeneration, could be automatically performed through a set procedure. After data processing, the detection results could be sent to the host server by a Wifi-based communication module.

### 2.3. Reusable Fiber Optic Bio-Probe Modified by Hapten-Carrier Protein Conjugate Using the DSC-Covalent-Coupling Strategy

Functionalization of the fiber optic bio-probe is essential for its sensitivity, specificity, and reusability. Using HF acid tube-etching method, the distal end of fiber optic was tapered to form a taper sensing region with length and diameter of 3.0 cm and 220 μm, respectively. We applied a novel modification method in our preparation of the fiber optic bio-probe (Figure 2). DON-BSA conjugates were covalently modified on the aminated biosensor surface using DSC as the bifunctional reagent, and there were regarded as the bio-recognition molecules. First, the combined tapered fiber optic bio-probe was prepared following our previously developed method. The bio-probe was initially placed in a piranha solution (H_2_O_2_:H_2_SO_4_ = 1:3) for 30 min to introduce hydroxyl groups onto its surface. After it was rinsed and dried in an oven at 105 °C, the bio-probe was immersed in an APTS toluene solution (2%, *v*/*v*) for 1 h to coat the reactive silane layer. After this was washed, the bio-probe was dried in an oven at 120 °C for 1 h to form a dense layer. The bio-probe was then placed in the DSC toluene solution (2%). After 2 h, the bio-probe was washed with ultrapure water and immersed in the 0.5 mg/mL DON-BSA solution for 12 h. The DON-BSA was covalently coupled to the bio-probe surface. Finally, the bio-probe was placed into a 2.0 mg/mL OVA solution for 2 h to block the non-specific adsorption sites. The bio-probe was stored at 4 °C for use.

### 2.4. Preparation of Real Samples and Their Detection Using the OIP-v2 and ELISA

Methanol was used for sample extraction in the real sample analysis. Quantities of 50–250 g of corn, wheat, or pig feed samples were collected. After passing through a 25-mesh screen, 1 g of each sample was weighed and a 5 mL methanol/water (8:2, *v*/*v*) solution was added as an extraction reagent. The mixture was shaken vigorously for 5 min and then centrifuged at 3000× *g* for 3 min. The supernatant was collected and spiked with DON from the DON stock solution. The supernatant was diluted 10 times using PBS solution before the analysis. After 50 μL of spiked samples was mixed with 50 μL Cy5.5-anti-DON antibody for pre-reaction, the mixture was introduced into the OIP-v2 to detect the DON concentration. Each spiked sample was tested in triplicate.

Nine naturally contaminated maize samples were also extracted and tested by this method. The supernatant was diluted 20 times using PBS before analysis. All of the samples were simultaneously analyzed using the enzyme-linked immunosorbent assay (ELISA) (Appendix A).

## 3. Results and Discussion

### 3.1. Characteristics of the DON-BSA Conjugates Modified Bio-Probe

Several experiments were conducted to evaluate the feasibility of the DON-BSA functionalized bio-probe for DON detection (see Appendix A online). First, 0.5 μg/mL Cy5.5-anti-DON antibodies were introduced into the optofluidic cell with an effective volume of 30 μL at a constant speed of 50 μL/min for 1 min, and the fluorescence signal detected by the OIP-v2 increased over time. A signal plateau was obtained after 5 min and the effective fluorescence signal (*F_e_*) was 204, which is calculated as follows:(1)$$F_{e}=F_{p}-F_{b}$$
where *F_p_* is the fluorescence intensity at the peak value and *F*_b_ is the baseline value. Second, a mixture of 0.5 μg/mL Cy5.5-anti-DON antibodies and 100.0 μg/L DON was introduced into the cell; the fluorescence signal also increased, but a lower fluorescence signal (*F_e_* = 120) was detected. Third, other antibodies, Cy5.5-anti-ENR antibody and Cy5.5-anti-MC-LR, were added, and few fluorescence intensities were observed even when their concentrations (1.0 μg/mL) were higher than that of anti-DON antibody (data not shown). These results demonstrate that the DON-BSA was successfully immobilized onto the bio-probe surface, and that the Cy5.5-labeled anti-DON antibodies modified bio-probe could be specifically bound with DON-BSA and were also inhibited by the addition of DON. Therefore, the functionalized bio-probe could be used for the DON immunoassay.

One of the main advantages of the OIP-v2, compared to other immunoassay methods (e.g., ELISA and colloidal gold-based immunoassay), is that it can be regenerated and reused, which is essential to ensure accurate detection, improve testing variation, and reduce costs. Figure 3 shows that the DON-BSA functionalized bio-probe could be regenerated after having been washed by the SDS solution (0.5%, pH = 1.9) and the PBS solution. After the bound Cy5.5-anti-DON antibodies were removed, the signal curve went back to the baseline. The DON-BSA functionalized bio-probe could be reused with more than 100 successive assays and few activity losses (<5%) (Appendix A). These results indicate that the binding properties of the DON-BSA modified onto the bio-probe could be retained after being washed by the SDS solution, and that the OIP-v2 was sufficiently reliable.

### 3.2. Dose-Response Curve of DON

The indirect competition immunoassay mechanism of DON using the OIP-v2 is shown in Figure 4a. Our detection process included several steps. First, 50 μL Cy5.5-anti-DON antibodies at a fixed concentration were mixed with 50 μL DON samples at different concentrations, and the mixture was pre-reacted for 5 min. Part binding sites of antibodies were specifically occupied by DON, which were proportional to the DON concentration. Second, the mixture of Cy5.5-anti-DON antibodies and DON samples was pumped into the optofluidic cell at a constant speed of 50 μL/min for 1 min and reacted for 4 min. Cy5.5-anti-DON antibodies with free binding sites bound with DON-BSA were modified onto the bio-probe surface, and the fluorescence intensity increased over time and was detected in real-time by the OIP-v2. Due to the limited penetration depth of the evanescent wave, free Cy5.5-anti-DON antibodies contributed less to the fluorescence intensity. Therefore, the washed step in other immunoassay methods (e.g., ELISA) was unnecessary. Finally, the bio-probe was regenerated using the SDS solution for 3 min, and washed using the PBS solution for 1 min. One detection process was less than 10 min.

Figure 4b,c shows the typical fluorescence signal traces for DON detection using the OIP-v2 when the concentrations of anti-DON antibodies were 0.5 μg/mL and 0.25 μg/mL, respectively. Increasing the concentration of DON resulted in a proportional decrease of the fluorescence intensity because fewer anti-DON antibodies bound with the DON-BSA onto the bio-probe surface. The fluorescence intensity of each DON concentration was normalized as the ratio to the fluorescence intensity of the blank sample containing no DON as follows:(2)$$I=\frac{F_{e,sample}}{F_{e,blank}}$$

The dose-response curves of DON were plotted against the logarithm of the DON concentration using a four-parameter logistic equation [20]. The error bars in Figure 4d were less than 6.7%, demonstrating good stability of the OIP-v2 for DON determination.

From Figure 4d, the LOD of DON, using three times the standard deviation of the mean blank value, was 0.16 μg/L (0.16 ppb) and 1.2 μg/L (1.2 ppb) when the Cy5.5-labeled anti-DON antibody concentrations were 0.25 μg/mL and 0.5 μg/mL, respectively. These LODs were by far lower than the limit value (1 ppm) set by China and the United States. The linear working ranges of DON were from 1.67 μg/L to 35.44 μg/L and from 0.43 μg/L to 36.61 μg/L when the anti-DON antibody concentrations were 0.5 μg/mL and 0.25 μg/mL, respectively. Increasing the antibody concentration increases the detection fluorescence intensity of the OIP-v2, but it might result in increasing the LOD and reducing the immunoassay detection range. Therefore, the antibody concentration should be optimized through experiments to improve immunoassay performance. The high sensitivity of the OIP-v2 for DON detection is comparable with most of the existing methods (e.g., HPLC-MS, other immunosensors) demonstrated for DON (details in Appendix A) [20,21,22,23,24,25,26]. There are several reasons for this response. First, the evanescent wave fluorescence immunoassay technologies have high sensitivity due to their low background noise. Second, the high sensitivity and signal-to-noise ratio of the OIP-v2 are beneficial for increasing immunoassay sensitivity. Third, the surface chemistry of the DON-BSA modified bio-probe surface kept its high activity toward the anti-DON antibodies [20]. Furthermore, the OIP-v2 manifested other advantages, including its simple system structure and portability, small sample volume (~30.0 µL), and the high reusability of the bio-probe with few activity losses for multiple immunoassays.

### 3.3. Selectivity of DON Immunoassay

The selectivity of an antibody is an essential analytical parameter regarding the reliability and specificity of an immunosensor [27]. If the antibody can be bound with cross-reactive substances presented in the samples, it will result in the erroneous detection of targets of interest. Therefore, selectivity of the antibody has to be assessed before applying an immunosensor to assay the food samples without the need of a separation process.

The selectivity of the anti-DON antibody is evaluated using several related compounds, including Enrofloxacin (ENR), Zearalenone (ZEN), Aflatoxin B1(AFB_1_), and OchratoxinA (OTA), replacing DON as competitors, which may present together with DON in real samples. As seen in Appendix A, the immunosensor exhibits high sensitivity toward DON without a significant response (<5% compared to the blank control) to the potential interferences tested. This selectivity should contribute to the selective binding of anti-DON antibodies to DON.

### 3.4. Effect of Extraction Solvent on the Immunoassay of DON

Methanol extraction plays an important role in the assay of pollutants in cereals [28,29]. Several studies have indicated that the immunoassays could be conducted in an organic solution of a certain concentration [30,31]. The rapid on-site immunodetection of DON is easily achieved if it can be detected directly after the methanol extraction. To do this, we investigated the effect of a methanol solution on the immunoassay of DON. Cy5.5-anti-DON antibodies, organic solvents, and DON of different concentrations were mixed. The final concentrations of methanol were 5%, 10%, and 20% (*v*/*v*). Here, we used a high concentration of anti-DON antibody, and its final concentration was 1.0 μg/mL. The assay process was similar to that described above. By increasing the concentration of methanol up to 20%, the fluorescence intensity detected by the OIP-v2 changed only very slightly when no DON was added to the samples (Appendix A). These results indicate that the binding reaction between the DON-BSA immobilized on a bio-probe surface and the Cy5.5-anti-DON antibodies were not affected by the methanol.

Figure 5 shows the dose-response curves of DON at different methanol concentrations. Various concentrations of DON were mixed with Cy5.5-anti-DON antibodies in the methanol/PBS solution. After incubation for 6 min, the mixture was pumped into the optofluidic cell, and fluorescence intensity was detected. The fluorescence intensity of each DON concentration was normalized as the ratio to the fluorescence intensity of the blank sample containing no DON, according to Equation (2). The dose-response curves of DON at a low concentration level of methanol were very similar to those of DON in the PBS solution. Thus, DON can be quantitative at low methanol concentrations. When the concentration of methanol was over 20%, the LOD of DON increased and the linear dynamic range of DON detection became small. Because methanol did not affect the binding between anti-DON antibodies and DON-BSA immobilized to a bio-probe surface, the binding between the anti-DON antibodies and DON in the solution should be, to some extent, inhibited. However, the normalized value was still linearly dependent on the DON concentration when the DON concentration ranged from 5.18 μg/L to 94.62 μg/L. Theoretically, the immunosensor can still be used to quantify the DON even if the methanol concentration increases to 20%. To avoid using more methanol, we chose 10% methanol for the next experiments.

### 3.5. Assay of Spiked Cereal Samples Using the OIP-v2

The interaction between antibody and antigen for immunoassay is generally affected by the matrix effect of the real samples (e.g., pH, ionic strength, and organic matters) and the extraction solvents. Therefore, evaluating the matrix effect is necessary for obtaining accurate and stable immunoassay methods of cereals (Table 1). Our recovery experiments of DON at different concentrations were carried out with spiked cereal samples, including corn, wheat, and pig feed. These samples were spiked with DON from the DON stock solution at concentrations of 5.0, 10.0, and 20.0 µg/L. The recoveries of the spiked samples were in the range from 81% to 127%, and the parallel test results show that the relative standard deviation (RSD) ranged from 3.92% to 10.46%. These are similar with the recoveries (77.1–107.0%) and RSD (4.2–11.9%) of the other immunological detection method in wheat [32]. These results indicate that the proposed method was capable of detecting DON in the real samples with suitable accuracy and precision, and the interference from the real samples was negligible.

### 3.6. Simultaneous Assay of Positive Samples Using the OIP-v2 and ELISA

The performance of the OIP-v2 was also evaluated through a comparison of the results obtained from the OIP-v2 using maize-positive samples with those obtained by ELISA (details in Appendix A). The maize samples were treated as described above, and simultaneously detected by the OIP-v2 and ELISA. The detection results and the contents of the nine maize samples, which range from 140 μg/mL to 2400 μg/mL, are shown in Appendix A. The DON concentration of four maize samples was higher than the Chinese regulatory limit of 1000 μg/kg (1 ppm). A comparison between the methods yielded good correlation results (y = 0.981x + 16.75) (Appendix A) and the correlation coefficient (R^2^ = 0.9891) demonstrates satisfactory agreement between the methods. These results indicate the adequacy of the OIP-v2 for accurate DON detection. The accuracy and reliability of the present method indicates that the developed immunoassay system can be successfully applied to DON analysis in cereal products using the methanol extraction method.

## 4. Conclusions

The improved portable and reusable OIP-v2 was developed for rapid and sensitive on-site detection of DON using DON-BSA modified bio-probes as biorecognition elements. The OIP-v2 was used for detection of DON with high sensitivity, accuracy, and rapidity. The LOD of DON was 0.16 µg/L under optimal conditions, which was far lower than the limit values set in both China and the United States. The recovery of DON spiked in cereal samples at various concentrations ranged from 81% to 127%. The immunoassay performance of the OIP-v2 was also validated with respect to ELISA through simultaneous detection of maize-positive samples, and the two methods were in good agreement. Compared with other biosensors, the OIP-v2 has several advantages. First, the robustness of the DON-BSA modified bio-probe surface allows multiple DON detection cycles without significant activity loss, resulting in more accurate and cost-effective results. Second, the limited penetrated depth of the evanescent wave may effectively discriminate free Cy5.5-anti-DON antibodies and bind Cy5.5-anti-DON antibodies onto the bio-probe surface, which greatly shortens the on-site detection time of DON. Third, the OIP-v2 can perform automated data processing and generate alarm signals when the concentration of pollutants exceeds a pre-set threshold value, and then send them directly to the food quality management department [31]. In addition, the measurement of the DON-spiked samples using the OIP-v2 showed good recovery, precision, and accuracy. Therefore, the OIP-v2 can serve as a powerful analytical tool for the on-site detection of other food pollutants only using their respective antibody and hapten-carrier protein-modified bio-probe, and can provide a reliable prognostication assessment of the food risk.

## Figures and Tables

**Figure 1 ijerph-19-03759-f001:**
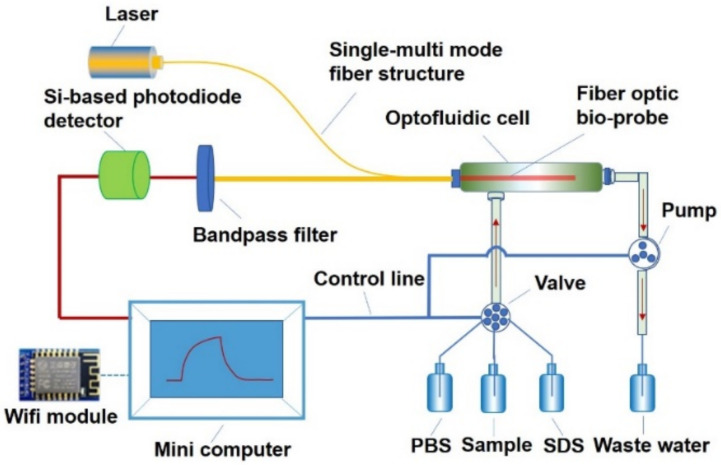
Scheme of the OIP-v2, including the optical system (laser, single-multi mode fiber structure, bandpass filter, and Si-based photodiode detector), the optofluidic system (optofluidic cells and fiber optic bio-probe), the fluidics (pump and valve), and a mini-computer.

**Figure 2 ijerph-19-03759-f002:**
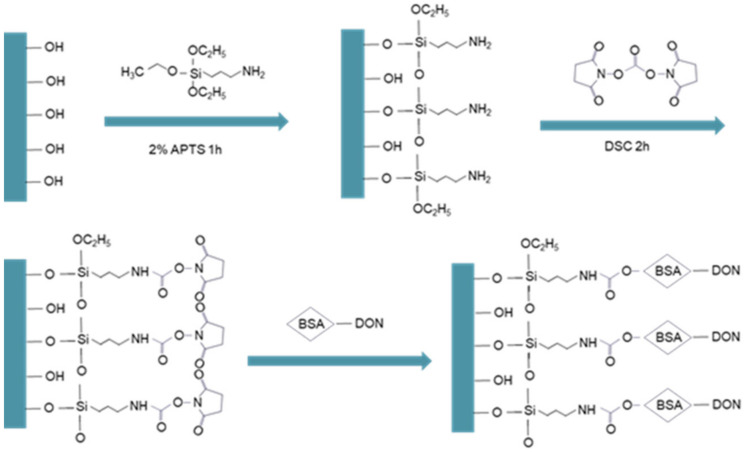
Preparation of the fiber optic bio-probe. DON-BSA conjugates were covalently modified on the aminated bio-probe surface using DSC as the bifunctional reagent.

**Figure 3 ijerph-19-03759-f003:**
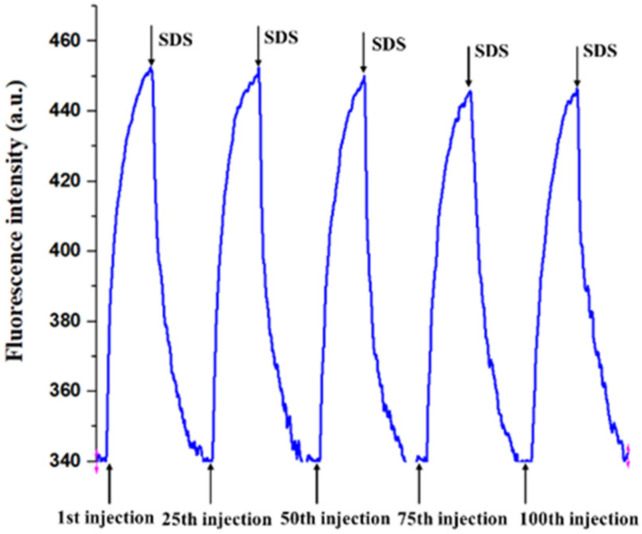
The fluorescence signal traces without DON using 0.25 μg·mL^−1^ Cy5.5-anti-DON-antibody for multiple immunoassays of the DON-BSA modified bio-probe.

**Figure 4 ijerph-19-03759-f004:**
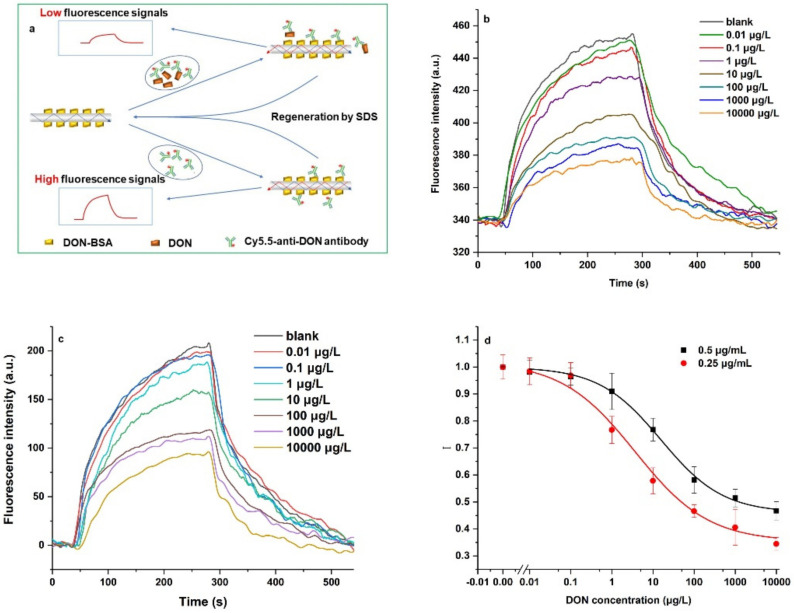
Detection of DON using the OIP-v2. (**a**) The indirect competition immunoassay mechanism of DON; (**b**) typical fluorescence signal traces with DON concentration ranging from 0 to 10,000 μg/L using 0.25 μg/mL Cy5.5-anti-DON-antibodies; (**c**) typical fluorescence signal traces with DON concentration ranging from 0 to 10,000 μg/L using 0.5 μg/mL Cy5.5-anti-DON-antibodies; and (**d**) the logistic-fitted dose-response curve for detection of DON. The error bars correspond to the standard deviation of the data points in the three repeated experiments (*n* = 3).

**Figure 5 ijerph-19-03759-f005:**
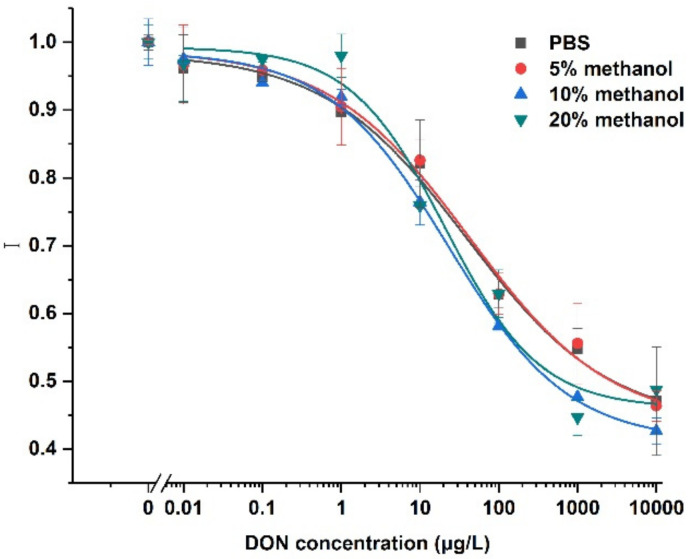
Dose-response curves for the detection of DON ranging from 0 to 10,000 μg/L at methanol concentrations of 0, 5%, 10%, and 20%. The error bars correspond to the standard deviation of the data points in each of the three repeated experiments (*n* = 3).

**Table 1 ijerph-19-03759-t001:** Spiked recovery of DON in the real samples (three replicates for each concentration).

Actual Sample	Spiked Concentration (μg/L)	OIP-v2		
		Detection (μg/L)	CV/%	RSD/%
Corn	5.0	5.88	117.66	3.99
	10.0	9.36	93.61	8.83
	20.0	21.11	105.56	4.13
Wheat	5.0	6.08	121.61	7.32
	10.0	8.05	80.50	7.78
	20.0	19.68	98.42	3.92
Pig Feed	5.0	4.57	91.44	7.83
	10.0	12.68	126.79	8.58
	20.0	21.41	107.05	10.46

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
