# Peer review of "Rapid, Sensitive On-Site Detection of Deoxynivalenol in Cereals Using Portable and Reusable Evanescent Wave Optofluidic Immunosensor"

_ijerph, 2022, doi:10.3390/ijerph19073759_

Round 1

Reviewer 1 Report

In the opinion of the Reviewer, Figure 3a adds nothing new to the article in this form, because the description is in the text of the study. The drawing should be redesigned to show the results and not the method of interpretation. Please modify the drawing or delete it.

Furthermore, the conclusions are too long and convoluted. Please shorten this fragment and indicate the specific dependencies and recommendations shown.

On behalf of the Reviewer, the following items could be added to the cited literature:

  1. J. Perkowski, K. Stuper, M. BuÅ›ko, T. Góral, H. JeleÅ„, M. Wiwart, E. Suchowilska, A comparison of contents of group A and B trichothecenes and microbial counts in different cereal species. Food Additives and Contaminants - Part B, 2012, 49,151-159
  2. BuÅ›ko, K. Stuper, H. JeleÅ„, T. Góral, J. Chmielewski, B. Tyrakowska, J. Perkowski, Comparison of volatiles profile and contents of trichothecenes group B, ergosterol, and ATP of bread wheat, durum wheat, and triticale grain naturally contaminated by mycobiota. FRONTIERS in Plant Science, 2016, 7, 1243-1251
  3. Stuper-Szablewska, J. Perkowski, Level of contamination with mycobiota and contents of mycotoxins from the group of trichotchecenes in grain of wheat, oat, barley, rye and triticale harvested in Poland in the years of 2006-2008, Annals of Agriculture Environment Medicine, 2017, 24(1), 49-55
  4. Kulik, K. Stuper-Szablewska, K. Bilska, M. Buśko, A. Ostrowska-Kołodziejczak, D. Załuski, J. Perkowski, Sinapic acid affects phenolic and trichothecene profiles of F. culmorum and F. graminearum sensu stricto. Toxins, 2017, 9, 264

Author Response

Question 1. In the opinion of the Reviewer, Figure 3a adds nothing new to the article in this form, because the description is in the text of the study. The drawing should be redesigned to show the results and not the method of interpretation. Please modify the drawing or delete it.

Response: Many thanks. Figure 3a showed the indirect competition immunoassay mechanism of DON. Although this was described in manuscript. we think the drawing make the reader to be easier to understand the detection process of DON. Therefore, we did not delete it.

Question 2. Furthermore, the conclusions are too long and convoluted. Please shorten this fragment and indicate the specific dependencies and recommendations shown.

Response: Thanks a lot. The conclusions have been shortened and rewritten.

Question 3. On behalf of the Reviewer, the following items could be added to the cited literature:

  1. J. Perkowski, K. Stuper, M. BuÅ›ko, T. Góral, H. JeleÅ„, M. Wiwart, E. Suchowilska, A comparison of contents of group A and B trichothecenes and microbial counts in different cereal species. Food Additives and Contaminants - Part B, 2012, 49,151-159
  2. BuÅ›ko, K. Stuper, H. JeleÅ„, T. Góral, J. Chmielewski, B. Tyrakowska, J. Perkowski, Comparison of volatiles profile and contents of trichothecenes group B, ergosterol, and ATP of bread wheat, durum wheat, and triticale grain naturally contaminated by mycobiota. FRONTIERS in Plant Science, 2016, 7, 1243-1251
  3. Stuper-Szablewska, J. Perkowski, Level of contamination with mycobiota and contents of mycotoxins from the group of trichotchecenes in grain of wheat, oat, barley, rye and triticale harvested in Poland in the years of 2006-2008, Annals of Agriculture Environment Medicine, 2017, 24(1), 49-55
  4. Kulik, K. Stuper-Szablewska, K. Bilska, M. Buśko, A. Ostrowska-Kołodziejczak, D. Załuski, J. Perkowski, Sinapic acid affects phenolic and trichothecene profiles of F. culmorum and F. graminearum sensu stricto. Toxins, 2017, 9, 264

Response: Thanks for your suggestions. Some of them have added in our manuscript.

Reviewer 2 Report

Dear Authors, this article is very important both from academic and practical results. There are some comments and general suggestions how you can improve this article.

By the end of introduction’s section, it is stated: “First, a novel modified method of the fiber optic bio-probe was proposed to improve its sensing performance in terms of specificity, stability and reusability.” Add some details: if the novelty of yours methods contents only modification or some more innovative solutions?

Technical parameters of used instruments and all apparatus need to be added in Figure 1 or below.

There are only comparison with local governmental norms in statements: “The DON concentration of four maize samples was higher than the Chinese regulatory limit of 1000 μg/kg (1ppm).” If you wish global wide range of international readers to yours results, there are more comparisons available from international requirements.

Conclusions could be improved by adding practical recommendations how we can use this new method in the similar research.

Sincerely, Reviewer.   

Author Response

Reviewer #2. This article is very important both from academic and practical results. There are some comments and general suggestions how you can improve this article.

Question 1. By the end of introduction’s section, it is stated: “First, a novel modified method of the fiber optic bio-probe was proposed to improve its sensing performance in terms of specificity, stability and reusability.” Add some details: if the novelty of yours methods contents only modification or some more innovative solutions?

Response: Many thanks for your comments. The stability and reusability of the DON-BSA functionalized bio-probe using DSC has been verified by our experimental results, which could be reused with more than 100 successive assays and few activity losses. However, the specificity of biosensor mainly originated from the biorecognition molecules. To avoid the confusion, we deleted the specificity. In this study, the hapten-carrier protein conjugates were covalently immobilized onto the bio-probe surface as biorecognition molecules using N, N′-disuccinimidyl carbonate (DSC) as the bifunctional reagent instead of other bifunctional reagents (e.g., glutaraldehyde), because the latter generally makes the bio-probe to have non-special adsorption due to its functional group according to our previous experience.

Question 2. Technical parameters of used instruments and all apparatus need to be added in Figure 1 or below.

Response: Many thanks for your suggestion. We have added the photo of instruments (Fig.S1) and its information.

Question 3. There are only comparison with local governmental norms in statements: “The DON concentration of four maize samples was higher than the Chinese regulatory limit of 1000 μg/kg (1ppm).” If you wish global wide range of international readers to yours results, there are more comparisons available from international requirements.

Response. Many thanks for your comments. In the introduction, we have mentioned “In China and the United States, the maximum permissible level of DON is 1 ppm in cereal and cereal products”.

Question 4. Conclusions could be improved by adding practical recommendations how we can use this new method in the similar research.

Response: Many thanks. We have revised the conclusions according to your suggestions.

Reviewer 3 Report

In this manuscript, the authors developed a competitive immunosensor for DON based on the evanescent wave fluorescence. The reviewer understands that the strong point of the present system is the unnecessity of the separation steps for unbounded antibody conjugated with fluorescence molecule (Cy5.5). However, this advantage has been already reported by authors (ref. 17). Although the authors described immunosensing platform was improved (line 63), the relationship between the improved points and their results was unclear. Important keywords (rapid, sensitive, on-site, potable and reusable) were employed in the title. However, it is quite complicated which improvements were connected to which keywords (characteristics). Therefore, the reviewer recommends major revision for the publication.

1. The authors adopted DSC (line 68) for immobilizing DON-BSA. Why this is good for the specificity, stability and reusability compared to other crosslinking agents? The authors should show us the results obtained by using DSC and others.

2. The authors used Si-based photodiode detector with high-sensitivity as an alternative to a PMT for the detection of a fluorescence signal. However, the readers cannot understand how sensitive it is.

3. The author described the system is potable. The size information and pictures for whole system should be added in the text. The incorrect figure was inserted for Fig. S1 (line 96-97).

4. The authors should move Fig. S1 to main text, if “reusable” is the improved point. The time information should be added in the horizontal axis.

5. There is contradiction for sample volume. Please confirm the difference between line 171 and 239.

6. The authors used SDS to remove antibody conjugated with Cy5.5. I guess that DON-BSA and OVA could also be removed by the treatment with surfactant.

7. Please add time information during introducing the solutions after each solution containing antibody and DON was mixed.

Author Response

Reviewer #3. In this manuscript, the authors developed a competitive immunosensor for DON based on the evanescent wave fluorescence. The reviewer understands that the strong point of the present system is the unnecessity of the separation steps for unbounded antibody conjugated with fluorescence molecule (Cy5.5). However, this advantage has been already reported by authors (ref. 17). Although the authors described immunosensing platform was improved (line 63), the relationship between the improved points and their results was unclear. Important keywords (rapid, sensitive, on-site, potable and reusable) were employed in the title. However, it is quite complicated which improvements were connected to which keywords (characteristics). Therefore, the reviewer recommends major revision for the publication.

Question 1. The authors adopted DSC (line 68) for immobilizing DON-BSA. Why this is good for the specificity, stability and reusability compared to other crosslinking agents? The authors should show us the results obtained by using DSC and others.

Response: Many thanks for your comments. The stability and reusability of the DON-BSA functionalized bio-probe using DSC has been verified by our experimental results, which could be reused with more than 100 successive assays and few activity losses. However, the specificity of biosensor mainly originated from the biorecognition molecules. To avoid the confusion, we deleted the specificity. In this study, the hapten-carrier protein conjugates were covalently immobilized onto the bio-probe surface as biorecognition molecules using N, N′-disuccinimidyl carbonate (DSC) as the bifunctional reagent instead of other bifunctional reagents (e.g., glutaraldehyde), because the latter generally makes the bio-probe to have non-special adsorption due to its functional group according to our previous experience.

Question 2. The authors used Si-based photodiode detector with high-sensitivity as an alternative to a PMT for the detection of a fluorescence signal. However, the readers cannot understand how sensitive it is.

Response: Thank you very much. The sensitivity of Si-based photodiode (SOP-1000) has been provided in our another paper (Song et al., Talanta 2019,196,78-84). We have added this reference. The sensitivity of SOP-1000could obtain 50 fW based on 3σ, and it was very suitable for low light intensity detection especially for fluorescence detection.

Question 3. The author described the system is portable. The size information and pictures for whole system should be added in the text. The incorrect figure was inserted for Fig. S1 (line 96-97).

Response: Many thanks for your comments. We have provided the photo of this system (Fig.S1), and the OIP-v2 has a size of 36×25×18cm and a weight of 2.5 kg.

Question 4. The authors should move Fig. S1 to main text, if “reusable” is the improved point. The time information should be added in the horizontal axis.

Response: Thank you very much. Fig.S1 have been moved to main text, it is now Figure 3. Because 100 successive assays needed to take a long time, we only took some typical processes to draw Fig.3 and labeled the testing times.

Question 5. There is contradiction for sample volume. Please confirm the difference between line 171 and 239.

Response: Many thanks. This is our typo, and has been revised as 30 µL.

Question 6. The authors used SDS to remove antibody conjugated with Cy5.5. I guess that DON-BSA and OVA could also be removed by the treatment with surfactant.

Response: Thanks a lot. In our method, the bio-probe was regenerated using the SDS solution for 3 min to remove the cy5.5-labeled antibody. However, The DON-BSA and OVA was covalently coupled to the bio-probe surface, and they can not be removed by SDS solution. This can be verified by high reusability of the DON-BSA functionalized bio-probe, which could be reused with more than 100 successive assays and few activity losses (Fig.3).

Question 7. Please add time information during introducing the solutions after each solution containing antibody and DON was mixed.

Response: Many thanks. We have added the time information in Section 3.2. The pre-treatment time of Cy5.5-anti-DON antibody and DON was 5 min.

Round 2

Reviewer 3 Report

In this revised manuscript, the authors appropriately revised the points raised by the reviewers. Hence, I suggest its acceptance for publication in the journal.